# Cellular Signal Transductions and Their Inhibitors Derived from Deep-Sea Organisms

**DOI:** 10.3390/md19040205

**Published:** 2021-04-05

**Authors:** Liyan Wang, Kazuo Umezawa

**Affiliations:** 1Shenzhen Key Laboratory of Marine Bioresource and Eco-Environmental Science, College of Life Sciences and Oceanography, Shenzhen University, Shenzhen 518060, China; lwang@szu.edu.cn; 2Molecular Target Medicine, School of Medicine, Aichi Medical University, Nagakute 480-1195, Japan

**Keywords:** cellular signal transduction, bioactive metabolite, deep-sea organisms, anti-inflammatory agent, anticancer agent

## Abstract

Not only physiological phenomena but also pathological phenomena can now be explained by the change of signal transduction in the cells of specific tissues. Commonly used cellular signal transductions are limited. They consist of the protein–tyrosine kinase dependent or independent Ras-ERK pathway, and the PI3K-Akt, JAK-STAT, SMAD, and NF-κB-activation pathways. In addition, biodegradation systems, such as the ubiquitin–proteasome pathway and autophagy, are also important for physiological and pathological conditions. If we can control signaling for each by a low-molecular-weight agent, it would be possible to treat diseases in new ways. At present, such cell signaling inhibitors are mainly looked for in plants, soil microorganisms, and the chemical library. The screening of bioactive metabolites from deep-sea organisms should be valuable because of the high incidence of finding novel compounds. Although it is still an emerging field, there are many successful examples, with new cell signaling inhibitors. In this review, we would like to explain the current view of the cell signaling systems important in diseases, and show the inhibitors found from deep-sea organisms, with their structures and biological activities. These inhibitors are possible candidates for anti-inflammatory agents, modulators of metabolic syndromes, antimicrobial agents, and anticancer agents.

## 1. Introduction

Nowadays most pathological phenomena such as inflammation, cancer, and diabetes mellitus can be explained by a change of signal transduction in the cells of specific tissues. Therefore, if we could control the specific signal transductions in the body, it would be possible to ameliorate diseases using new concepts. There are several ways to control cellular signal transductions; they are gene-editing therapy; protein therapy, such as providing antibodies or growth factors; and chemotherapy using low molecular weight compounds. Among them, chemotherapy has several advantages, because it is ready to use and free from ethical problems and unwanted immunity. Moreover, signal transduction inhibitors of low molecular weight can be screened similarly to antibiotics, anticancer agents, and enzyme inhibitors from nature.

In the present review, we explain signal transductions in relation to diseases, the methodology of inhibitor screening, screening sources past and present, and finally, we describe examples of the isolation of cell signaling inhibitors from deep-sea organisms.

## 2. Cellular Signal Transduction and Alteration in Disease

Cyclic AMP was discovered by Sutherland in the 1950s and was called a “second messenger” [1]. Since then, many second messengers that are intracellular signaling transducers have been found. A typical image of a signal transduction from the extracellular ligand to transcription factors is shown in Figure 1A. The signaling molecules shown by the arrow are enzymes or signaling molecules without enzyme activity. The most common signal transductions for the activity of various ligands, hormones, and growth factors are shown in Figure 1B. It is interesting that only a limited number of signaling pathways are commonly used for most activities in the body, even though there are large numbers of tissues, cell types, and functions.

Many growth factors, such as PDGF, EGF, VEGF, HGF, and IGF, bind to the cell surface receptors having protein–tyrosine kinase to activate the Ras-ERK and PI3K-Akt pathways (Figure 1B, R_1_). The growth promoting Ras-ERK pathway and apoptosis-inhibiting PI3K-Akt pathway are considered to be suitable targets for anticancer agents. Both the surface receptor-type, and cytoplasmic type, of protein–tyrosine kinases are associated with many oncogene products. We isolated a novel protein–tyrosine kinase inhibitor, lavendustin, from *Streptomyces* [2]. Bcr-Abl protein–tyrosine kinase is a cytoplasmic type, and the enzyme activity is activated in chronic myelogenous leukemia. Its inhibitor, imatinib, is a successful example of a signal transduction inhibitor that has been developed as orally active anticancer agent against chronic myelogenous leukemia [3]. Protein–tyrosine phosphatase (PTPase) is an opposite enzyme of protein–tyrosine kinase, and known to weaken the insulin-dependent signal transduction. Therefore, its inhibitors may become drugs for type 2 diabetes mellitus. We isolated the first naturally occurring PTPase inhibitor, dephostatin, from *Streptomyces* [4], and its designed analog was shown to inhibit PTP-1B and lower blood glucose in vivo [5].

Receptors of many amino acids or peptide hormones, such as adrenalin, glucagon, vasopressin (VP), oxytocin (OX), gonadotropin releasing hormone (GnRH), ACTH, and enkephalin, activate receptors (Figure 1B, R_2_) to activate either cyclic AMP production or phospholipase C activity. This cyclic AMP pathway was discovered by Sutherland [1].

Receptors of growth hormone (GH), erythropoietin (EP), and obesity-preventing leptin (Figure 1B, R_3_) do not possess protein–tyrosine kinase, but they activate the kinase activity of JAK in order to activate the STAT transcription factor activity.

TNF-α, interleukin (IL)-1β, and lipopolysaccharide (LPS) bind to the receptors that activate NF-κB (Figure 1B, R_4_). Typically, each receptor activates either TRAF6 or TRAF2, which then activates I-κB kinase (IKK) to facilitate the degradation of endogenous inhibitor I-κB by the ubiquitin/proteasome system. NF-κB is a transcription factor that enhances expressions of many inflammatory cytokines and anti-apoptosis proteins. NF-κB is often over-activated in tissue with inflammation and also in cancer cells. Therefore, NF-κB is an attractive target for chemotherapeutic agents, although it is also essential for physiological processes such as blood cell differentiation. The TNF-α antibody is now being used for anti-inflammatory therapy in rheumatoid arthritis and inflammatory bowel disease. Dehydroxymethylepoxyquinomicin (DHMEQ) was discovered by one of the authors as a specific inhibitor of NF-κB. It is a designed compound based on the structure of weak but non-toxic antibiotic epoxyquinomicin [6,7]. It is being developed as an anti-inflammatory ointment for atopic dermatitis and severe skin inflammation [8]. Avoiding the side effects of NF-κB inhibitors, DHMEQ intraperitoneal therapy is being developed for the suppression of cancer [9].

TGF-β binds to cell surface receptors (Figure 1B, R_5_) to activate SMAD proteins acting as transcription factors. It often induces fibrosis in the liver, lungs, pancreas, and peritoneal cavity. TGF-β also accelerates the epidermal mesenchymal transition (EMT) that enhances malignancy in cancer cells [10]. We found that the plant-derived alkaloid conophylline inhibited TGF-β receptor downstream signaling [11].

The Wnt/β-catenin pathway is often activated in cancer cells. In this pathway, stimulation of the receptor by the Wnt ligand (Figure 1B, R_6_) activates GSK-3β to activate β-catenin. Aggregated β-catenins enter into the nucleus to act as a transcription factor together with the TCF–LEF complex.

The calcineurin–NFAT pathway was discovered by a chemical biology technique using a fishing probe of immune-suppressant FK506 (tacrolimus) derivative [12]. After the incorporation of calcium ions through the calcium channel (Figure 1B,C), calcium ion-dependent calmodulin activates calcineurin to activate the nuclear factor of activated T-cells-3 (NFAT3). FK506 and cyclosporine inactivate calcineurin through their binding proteins.

Steroid hormones are lipophilic and can pass through the cell membrane (Figure 1C). Their receptors exist in the cytoplasm and after binding the ligand-receptor complex enter the nucleus to bind to the promoter site of DNA, acting as a transcription factor. Liver X receptors also exist in the cytoplasm and their ligands bind to the receptor there to enter the nucleus.

In addition to the signal transductions in Figure 1B,C, there are two protein degradation pathways that have been recently developed. The ubiquitin–proteasome pathway is important for the regulation of cellular signal transduction [13]. For example, activation of NF-κB requires degradation of inhibitory protein I-κB, and the degradation is carried out by proteasomes after ubiquitination (Figure 2A). Bortezomib is an inhibitor of proteasomes and is now being clinically used for the treatment of multiple myeloma [14]. While the ubiquitin–proteasome system is important for the degradation of specific proteins, autophagy is important for the degradation of mass biomaterials. The process of autophagy (Figure 2B) is initiated by the formation of autophagosome during the wrapping of biomaterials by the double membrane system in the cytoplasm [15]. Autophagy is caused by the lysosome-mediated degradation of damaged proteins and organelles, as well as unwanted bacteria and viruses. It is essential for the loss of mitochondria in the evolution of red blood cells and for the cellular antibacterial activity against *Mycobacterium tuberculosis* [16]. The age-dependent decrease in autophagy activity in the brain is considered to accelerate neurodegenerative diseases. We have looked for the activators of cellular autophagy and have previously reported that the plant-derived alkaloid conophylline ameliorated cellular models of Parkinson’s and Huntington’s diseases by the activation of autophagy [17]. Conophylline also activated autophagy in the liver of high fat diet-induced non-alcoholic steatohepatitis (NASH) mice [18]. Although molecular targets for the screening of autophagy activators or inhibitors are not yet clear, it would be useful to look for the non-toxic regulator of autophagy.

Thus, not only the physiological role of tissues and cells, but also the mechanism of diseases can be explained by changes of signal transductions, mostly shown in Figure 1 and Figure 2. The chemical inhibitors of cellular signal transduction, which is over-activated in the cells or tissues in diseases, should be useful as a rational drug for each disease.

## 3. Process of Screening and Deep-Sea Organisms as a Source of Bioactive Metabolites

The process of bioactive metabolite screening we employed is shown in Figure 3A. Setting up a biological assay system is the most important process. The biological assay systems include antibacterial activity, measurement of enzyme activity, and cellular morphology, growth, apoptosis, and differentiation. The biological activity should reflect the essential signal transduction in diseases. In the case of drug screening, if the activity is not essential for the etiology of disease, the screened signaling inhibitor will be useless in further disease models or at the clinical stage. The biological activity should be simple and carried out economically and effectively. The enzyme assay and cellular assay are the most popular for screening.

After the screening of plants or microorganisms to find a hit, we began isolation using various chromatography techniques and structure determination by spectroscopy, including nuclear magnetic resonance and mass spectrometry. After determining the structure, we studied the effect of signal transduction inhibitors on various biological activities, including the cellular and animal toxicity. If it is possible, the compound is synthesized to facilitate enough supply and derivative preparation. If the original or derivative compound suppresses the disease models satisfactorily, we will try to develop it into a drug together with the industry. Meanwhile, we have been studying the mechanism of diseases using the signal transduction inhibitors in cultured cells or animal experiments.

Secondary metabolites are defined as low-molecular weight compounds produced by biological organisms that are not essential for the life of the producers. Secondary metabolites are produced only by plants and microorganisms. Tetrodotoxin is produced by fish and many secondary metabolites are produced by sponges. However, these compounds are considered to be produced by parasitic microorganisms. Bioactive metabolites are secondary metabolites that possess specific biological activities. The specific biological activity should be shown at comparatively low concentrations or doses.

Penicillin was isolated from a fungus, *Penicillium notatum*, and streptomycin from *Streptomyces griseus*, in the 1940s. Since then, many antibiotics, anticancer drugs, and enzyme inhibitors have been isolated from microorganisms and plants. Microorganisms and plants are still useful sources for the screening of useful drugs. Natural sources of signal transduction inhibitors are summarized in Figure 3B.

Many scientists have tried to isolate novel compounds from plants, soil microorganisms (including bacteria, *Streptomyces*, and fungi), and ordinary marine organisms since the middle of the 20th century. However, after a long history of screening, it is getting more difficult to find novel compounds anywhere in the world.

Meanwhile, more than 28,600 marine natural products have been reported. However, with the development of marine natural product research, the hit rate of new compounds is decreasing. Therefore, scientists are turning their attention to the deep sea. By 2008, almost 400 compounds were isolated from deep-sea organisms. By 2013, a further 188 new deep-sea natural products had been reported. About 75% of compounds from such an origin were reported to show biological activity (i.e., 141 of 188 compounds), with almost half (i.e., 81 of 188 compounds) exhibiting potent cytotoxicity in human cancer cell lines [19]. Blunt reported the effective screening of cytotoxic compounds with double the frequency from a single deep-water collection in New Zealand, compared with that of shallow water collections [20].

Sponges, corals, and fish also produce bioactive metabolites, although these metabolites are considered to be produced by parasitic microorganisms. In addition to the microorganisms, there are also sponges and corals in the deep sea. An investigation of the extracts of 65 “twilight zone” (50–1000 m depth) sponges, gorgonians, hard corals, and sponge-associated bacteria resulted in an extremely high hit rate (42%) of active extracts, with that for sponge and gorgonian extracts being 72% [21,22]. Thus, deep-sea organisms are considered to be important sources of natural products, especially for the screening of new cell signaling inhibitors.

## 4. Anti-Inflammatory Agents from Deep-Sea Organisms

### 4.1. Cyclopenol and Cyclopenin Inhibiting NF-κB Signaling

In the course of screening lipopolysaccharide (LPS)-induced nitric oxide (NO) production inhibitors, two related benzodiazepine derivatives, cyclopenol and cyclopenin (Figure 4), were isolated from the extract of a fungal strain, *Aspergillus* sp. SCSIOW2 [23]. The fungus was isolated from a deep marine sediment sample collected in the South China Sea at a depth of 2439 m. Cyclopenin was first isolated and reported in 1954 from a strain of *Penicillium cyclopium* [24], and cyclopenol was then isolated in 1963 from the same strain [25]. Cyclopenol and cyclopenin inhibited the LPS-induced formation of NO and secretion of interleukin-6 (IL-6) in RAW264.7 cells at non-toxic concentrations. In terms of the mechanism underlying these effects, cyclopenol and cyclopenin were found to inhibit the upstream signal of NF-κB activation.

Microglia are located in the brain for cleaning and protection against microorganisms, acting as macrophages in the brain. The 6-1 mouse microglia cell line was established and used to examine whether cyclopenol and cyclopenin inhibit LPS-induced inflammatory mediator protein expression in microglia. The results revealed that both clearly inhibited the expression of interleukin-1β (IL-1β), IL-6, and inducible NO synthase (iNOS).

Finally, the anti-inflammatory effects of cyclopenol and cyclopenin were examined in vivo. The ameliorative effect on learning deficits was assessed using amyloid-β-overexpressing *Drosophylla* flies, since only a small quantity of sample was necessary in this assay. Memantine, a derivative of adamantane, is an NMDA receptor antagonist clinically used for the treatment of Alzheimer’s disease. Memantine was used as the positive control. We found that cyclopenin rescued learning impairment, similar to memantine [23]. By contrast, cyclopenol did not ameliorate learning activity impairment. The hydroxyl group in cyclopenol may reduce the permeability of the compound and would make penetration into the body difficult.

### 4.2. Myrothenols Inhibiting LPS-Induced NO Production

We recently looked for novel compounds having anti-inflammatory activity from deep-sea microorganisms. As a result, we have isolated four new compounds, a pair of 2-benzoyl tetrahydrofuran enantiomers, named (−)-1*S*-myrothecol (Figure 4), (+)-1*R*-myrothecol (Figure 4), methoxy-myrothecol, and an azaphilone derivative, myrothin, from the culture filtrates of the deep sea-derived fungus, *Myrothecium* sp. BZO-L062 [26]. The fungus was isolated from a deep marine (2130 m deep) sediment sample, collected from an area close to Yongxing Island, China. The enantiomeric (−)-1*S*- and (+)-1*R*-myrothecol were separated by chiral normal phase high performance liquid chromatography (HPLC). Among all the isolated compounds, (−)-1*S*- and (+)-1*R*-myrothecol showed cellular anti-inflammatory activity inhibiting NO formation in LPS-treated macrophage-like cells.

LPS-induced NO production in mouse monocytic leukemia RAW264.7 cells, which are often employed for the evaluation of cellular anti-inflammatory activity because of similarities in phenotypes with macrophages. Both (−)-1*S*- and (+)-1*R*-myrothecol inhibited the LPS-induced NO production at non-toxic concentrations. The mechanism of inhibition remains to be elucidated.

Anti-oxidant activities can be measured by oxygen radical absorbance capacity (ORAC) assay. Both (−)-1*S*- and (+)-1*R*-myrothecol showed an antioxidant activity in the ORAC assay, with EC_50_ of 1.20 and 1.41 µg/mL, respectively, which were comparable with the positive controls, l-ascorbic acid (EC_50_, 1.55 µg/mL) and trolox (EC_50_, 1.61 µg/mL).

### 4.3. Macrolactins Inhibiting NO and Cytokine Productions

Novel macrolactin, 7,13-epoxyl-macrolactin A (Figure 4), was isolated from deep-sea sediment as an inhibitor of LPS-induced inflammatory mediator expression in RAW264.7 cells [27]. The producing strain was isolated from sediment collected at a depth of 3000 m in the Pacific Ocean, and identified as *Bacillus subtilis B5* by the complete 16S rRNA gene sequence. This new macrolactin exhibited a more potent inhibitory effect on NO production and several inflammatory cytokines than the previously known macrolactins, such as macrolactin A and analogues. Macrolactin A also inhibited the mRNA expression of iNOS, IL-1β, and IL-6 in LPS-stimulated RAW 264.7 cells.

### 4.4. Acremeremophilanes Inhibiting LPS-Induced NO Production

Separation and structural determination of an EtOAc extract of the culture filtrate of the fungus, *Acremonium* sp., from deep-sea sediment, resulted in the isolation of 15 new eremophilane-type sesquiterpenoids, acremeremophilanes A–O [28]. The fungus was collected from sediment at a depth of 2869 m in the South Atlantic Ocean.

All compounds were evaluated for inhibitory effects toward LPS-induced NO production in RAW 264.7 cells. Among them, acremeremophilane B (EC_50_, 8 µM, Figure 4) and E (EC_50_, 15 µM, Figure 4) showed comparatively potent inhibitory activities at non-toxic concentrations. A positive control, quercetin, inhibited NO production with EC_50_ of 15 µM in this system. The mechanism of inhibition remains to be elucidated.

### 4.5. Eutyperemophilanes Inhibiting LPS-Induced NO Production

Anti-inflammatory agents were searched for in the manipulated deep-sea microorganisms. The fungus, *Eutypella* sp. MCCC 3A00281, was collected from sediment at the extreme depth of 5610 m in the South Atlantic Ocean. Cultivation of the fungus by chemical epigenetic manipulation using suberohydroxamic acid, a histone deacetylase inhibitor, resulted in a significant change in the metabolite profile. Chromatographic application of the extended metabolites led to the isolation of a total of 30 eremophilane-type sesquiterpenoids, of which 26 were identified as new compounds, namely eutyperemophilanes A–Z [29]. All the compounds were evaluated for inhibitory effects on LPS-induced NO production in RAW264.7 cells. Among the 30 compounds, eutyperemophilane I and J (Figure 4) showed comparatively potent inhibition with IC_50_ of 8.6 and 13 µM, respectively (positive control quercetin, 16 µM), all at nontoxic concentrations.

### 4.6. Chrysamide C Inhibiting Interleukin-17 Production

Three dimeric nitrophenyl trans-epoxyamides, chrysamides A–C, were obtained from the deep-sea-derived fungus, *Penicillium chrysogenum* SCSIO41001 [30]. The fungal strain used was isolated from deep-sea sediment from the Indian Ocean at a depth of 3386 m. These compounds showed cytotoxicity at 30 µM in cancer cells. Chrysamide C (Figure 4) only possessed the oxazolidine ring, differentiating it from chrysamide A and B. Chrysamide C, but not A and B, suppressed the production of proinflammatory cytokine interleukin-17 (IL-17). The inhibitory rate on the production of IL-17 was 40.06% at 1.0 μM, while the rate of the positive control SR2211 was 62.86% at 1.0 μM. For the evaluation of IL-17 production, naive T-cells were isolated from IL-17-GFP reporter mice spleen, and then stimulated with anti-CD3, anti-CD28, TGF-β, IL-6, anti-IFN-γ, and anti-IL-4 in the presence of test chemicals. IL-17 is mainly produced by activated T-lymphocytes, and induces inflammatory cytokine and chemokine secretions in fibroblasts, epithelial cells, endothelial cells, and macrophages.

### 4.7. Butyrolactone I Suppressing Mast Cell Activity

A food allergy is defined as an immune-mediated adverse reaction to a food. Egg allergy is common in children worldwide and is considered to be mainly caused by ovalbumin. Therefore, ovalbumin is often used to construct animal models of food allergy. A type I allergic reaction is mediated by antigen specific IgE that causes mast cell degranulation [31], and this reaction leads to anaphylactic shock when it takes place in the whole body [32]. Mast cells secrete leukotrienes, histamine, and prostaglandins upon activation with antigen and IgE, and they play a central role in allergic reactions. Allergic inflammation is characterized by the tissue infiltration of inflammatory cells, including mast cells, macrophages, and lymphocytes. For the cellular assay of an allergic reaction, antigen and IGE-responsive rat basophilic leukemia RBL-2H3 cells are often employed [33].

Butyrolactone I (Figure 4), which was identified as a new type of butanolide, was isolated from a deep-sea-derived fungus, *Aspergillus* sp. [34]. The fungus was isolated from a hydrothermal sulfide deposit in the southwest Indian Ocean at a depth of 2783 m.

Ovalbumin-induced BALB/c mouse anaphylaxis model was established to study food allergic activity. Butyrolactone I ameliorated ovalbumin-induced allergy symptoms, and reduced the levels of histamine and mouse mast cell proteinases. It inhibited ovalbumin-caused production of IgE, and inhibited the accumulation of mast cells in the spleen and mesenteric lymph nodes. It also significantly suppressed mast cell-dependent passive cutaneous anaphylaxis. Additionally, the butyrolactone-I caused down-regulation of c-KIT receptors to reduce maturation of mast cells. Moreover, molecular docking analyses revealed that butylolactone I would interact with the inhibitory receptor, FcγRIIB.

### 4.8. Reticurol Suppressing Mast Cell Activity

The same group carried out screening of bioactive compounds from the hydrothermal fungus, *Graphostroma* sp. MCCC 3A00421 [35]. The fungus was isolated from deep-sea hydrothermal sulfide deposits from the Atlantic Ocean at a depth of 2721 m. Nine new compounds, including graphostrin A, and 19 known polyketides, were isolated. All the isolated compounds were tested for cellular anti-food allergic bioactivity in antigen and IgE-treated RBL-2H3 cells. Among them, reticulol (Figure 4), a known polyketide, effectively decreased the rates of degranulation and histamine release, with IC_50_ values of 13.5 and 13.7 μM, respectively. Reticulol was first isolated by Hamao Umezawa and co-workers from *Streptomyces mobaraensis* in 1977 as an inhibitor of cyclic nucleotide phosphodiesterase [36]. It was later discovered to inhibit topoisomerase I [37] in addition to cyclic nucleotide phosphodiesterase, and was shown to exhibit anticancer activity in vivo [38].

## 5. Modulators of Metabolic Syndrome Model and Antimicrobial Compounds

### 5.1. Puniceloids C and D, Liver X Receptor Agonists

Liver X receptors (LXR), including LXRα and LXRβ, are critical modulators of cholesterol and lipid metabolism, inflammatory responses, and innate immunity [39]. LXRs are ligand-activated transcription factors that belong to a family of hormone nuclear receptors (Figure 1C). LXR agonists have been suggested to have a potential use in the treatment of atherosclerosis, diabetes, inflammation, and Alzheimer’s disease [40].

Eight novel diketopiperazine-type alkaloids, including four oxepin-containing diketopiperazine-type alkaloids, oxepinamides H–K, and four 4-quinazolinone alkaloids, puniceloids A–D, were isolated from culture broth extracts of the deep-sea-derived fungus, *Aspergillus puniceus* SCSIO z021 [41]. The fungus was isolated from deep-sea sediment collected in the Okinawa Trough at the depth of 1589 m, about 4.7 km away from active hydrothermal vents. For the measurement of LXR agonist activity, human hepatocyte L02 cells were transfected by the LXRα reporter system with luciferase gene. These eight compounds showed significant transcriptional activation of LXRα. Among them, puniceloid C and D (Figure 5) were the most potent, both with EC_50_ values of 1.7 μM.

### 5.2. Chrysopyrones A and B, Protein–Tyrosine Phosphatase Inhibitors That Ameliorate Diabetes Mellitus Model

Diabetes mellitus is caused by two factors: a decrease in insulin production in the pancreatic islet, and a decrease in insulin sensitivity in the target tissues, such as muscle, fat, and liver. The latter is caused by a reduction of insulin-dependent signaling pathways, including insulin receptor, insulin receptor substrates (IRS), phosphatidylinositol 3-kinase (PI3K), and AKT (also called protein kinase B or serine/threonine kinase 1). The insulin receptor possesses protein–tyrosine kinase activity, and its substrates are the insulin receptor itself and IRS. Tyrosine phosphorylation of these proteins is essential for insulin signal transduction. However, if there is protein–tyrosine phosphatase in the cells, this enzyme removes phosphate from the tyrosine residues to weaken the insulin signal. Therefore, inhibitors of protein–tyrosine phosphatase, especially protein–tyrosine phosphatase-1B (PTP1B), should reactivate the insulin-dependent signaling to show the anti-diabetic activity [5,42].

Two new 3,4,6-trisubstituted α-pyron compounds, chrysopyrones A and B (Figure 5), as well as another new compound, penilline C, and 12 known compounds were isolated from the products of the fungus *Penicillium chrysogenum* SCSIO 07007 [43]. This fungus was separated from deep-sea hydrothermal vent environment samples collected from the western Atlantic at a depth of about 1000 m. Hydrothermal vents are formed through rock fissures located in volcanic regions. Deep-sea hydrothermal vent areas are characterized by high concentrations of reduced sulfur compounds.

Among the isolated compounds, chrysopyrones A and B (Figure 5) inhibited PTP1B with IC_50_ values of 9.32 and 27.8 μg/mL, respectively. They did not show cytotoxicity in cultured cells at 100 μg/mL. Their in vivo anti-diabetic activity remains to be studied.

### 5.3. Fiscpropionate A and C Inhibiting Bacterial Protein–Tyrosine Phosphatase

It is interesting that protein–tyrosine phosphatase also exists in bacteria and is considered to be the molecular target of antibacterial agents. In particular, *Mycobacterium tuberculosis* protein–tyrosine phosphatase B (MptpB) is an important virulence factor secreted by *Mycobacterium tuberculosis* into the host cell [44,45]. MptpB removes phosphate from the tyrosine residue of host proteins that are involved in the host signaling pathways, and it can attenuate the host immune defenses against tuberculosis. Therefore, MptpB inhibitors can enhance host immunity against *Mycobacterium tuberculosis*.

Fiscpropionates A–F, six new polypropionate derivatives featuring an unusual long hydrophobic chain, were isolated from the deep-sea-derived fungus, *Aspergillus fischeri* FS452 [46]. The fungal strain was isolated from deep-sea sludge in the Indian Ocean at a depth of 3000 m. Fiscpropionates A–D exhibited significant inhibitory activities against MptpB. Among them, fiscpropionates A and C (Figure 5) were comparatively effective, with the IC_50_ values of 5.1 and 4.0 μM, respectively. These compounds may be unique seeds of anti-tuberculosis agents.

### 5.4. Spiromastilactone D Inhibits Influenza Virus Replication

A new class of phenolic lactones with the trivial names of spiromastilactones A–M was isolated from a deep-sea-derived fungus, *Spiromastix* sp. MCCC 3A00308 [47]. The fungus was isolated from sediment collected from the South Atlantic Ocean at a depth of 2869 m. The structures feature varied chlorination of the aromatic rings. An antiviral assay revealed that most of the tested compounds exerted inhibitory activity against influenza virus replication in vitro at nontoxic concentrations. Among them, spiromastilactone D (Figure 5) showed the most potent activity for inhibiting a panel of influenza A and B viruses, in addition to drug-resistant clinical isolates. A mechanistic study using surface plasmon resonance suggested that the molecular target of spiromastilactone D would be hemagglutinin. Hemagglutinin is located in the envelope of the virus, and spiromastilactone D is likely to disrupt the interaction between hemagglutinin and the host sialic acid receptor, which is essential for the attachment and entry of all influenza viruses. In addition, spiromastilactone D showed inhibitory effects toward viral genome replication via targeting viral RNP complex. The synergistic effects on both viral entry and replication indicated it to be a candidate new anti-influenza agent.

## 6. Anticancer Agents

### 6.1. Cytotoxic Agents and Cell Signaling Inhibitors

Major anticancer drugs such as cisplatin, 5-fuluorouracil (5FU), doxorubicin, vinblastine, and paclitaxel are all cytotoxic compounds. Cisplatin, 5FU, and doxorubicin interact with DNA to inhibit polynucleotide synthesis, while vinblastine and paclitaxel bind to tubulin to inhibit cellular mitosis. These anticancer drugs attack not only cancer cells, but also normal cells, causing side effects. Although several new cytotoxic agents have been isolated from deep-sea organisms, they are not shown in this review. In general, the molecular targets of cytotoxic agents are difficult to find.

Meanwhile, the characteristics of cancer cells include fast growth rate, immortality, less requirement of growth factors, ability for anchorage-independent growth, less contact inhibition of growth, and so on. The selectivity of these anticancer drugs is found only in the fast growth of cancer cells. Generally, these anticancer agents are more effective in suppressing the growth of actively growing cells. Therefore, they damage normal, fast growing cells, such as bone marrow cells, intestine epithelial cells, skin cells, and reproductive organ cells, inducing serious side effects. Imatinib [3] and all-trans retinoic acid (ATRA) [48] are exceptional, since they show anticancer activity through the attack of molecular targets of cancer cells. These molecular target medicines inhibit specific cell signaling pathways.

### 6.2. Salinosporamide A, a Proteasome Inhibitor

The ubiquitin proteasome pathway was discovered in the 1980s, and it is now one of the most important cellular protein-degradation machineries. This pathway is essential for the removal of misfolded proteins, and it is also essential for the regulation of cell cycle and apoptosis [49]. The ubiquitin proteasome pathway down-regulates cell-cycle and tumor-suppressor proteins, such as p21, p27 [50], and p53 [51]. On the other hand, this pathway up-regulates oncogenic proteins, including NF-κB, by the degradation of inhibitory proteins [14]. Bortezomib is a successful example of proteasome inhibitors that are widely used clinically for the treatment of multiple myeloma [14].

Salinosporamide A (Figure 6) was isolated by Fenical et al. in 2003 from the marine actinomycete *Salinispora tropica*, collected at a depth of 1100 m, and it is a highly potent and selective inhibitor of the 20S proteasome [52]. It is a pyrrolidinone compound fused to a beta-lactone, and the structure is partly related to that of lactacystin (Figure 6) [53], an inhibitor of proteasome, isolated from *Streptomyces*. The crystal structure of salinosporamide A-20S proteasome revealed the mechanism of irreversible inhibition with β-lactone ring opening [54].

It suppressed both constitutive and inducible NF-κB activity [55]. Compared with bortezomib and lactacystin, salinosporamide A was found to be the most potent suppressor of NF-κB. Salinosporamide A inhibited I-κBα degradation, nuclear translocation of p65, and NF-κB-dependent gene expression in TNF-α-treated cells, while it showed no effect on I-κB kinase. Clinical trials of salinosporamide A are ongoing for the treatment of multiple myeloma.

## 7. Conclusions and Perspective

Cell signaling inhibitors from deep-sea organisms are summarized in Table 1.

Commonly used cellular signal transductions, such as protein–tyrosine kinase-dependent or independent Ras-ERK pathways and NF-κB-activation pathways, the ubiquitin–proteasome pathway and autophagy, are attractive targets for the discovery of safe, anti-inflammatory and anticancer drugs. At present, such cell signaling inhibitors are being looked for mainly in plants, soil microorganisms, and the chemical library. The screening of bioactive metabolites from deep-sea organisms is attractive because of the high incidence of finding novel compounds. Although it is still an emerging field, there are many successful examples of new cell signaling inhibitors. In particular, a proteasome inhibitor, salinosporamide A, isolated from deep-sea microorganisms is being developed as an anticancer agent. In the future, there are likely to be more cell signaling inhibitors from the deep-sea that will be developed into drugs.

## Figures and Tables

**Figure 1 marinedrugs-19-00205-f001:**
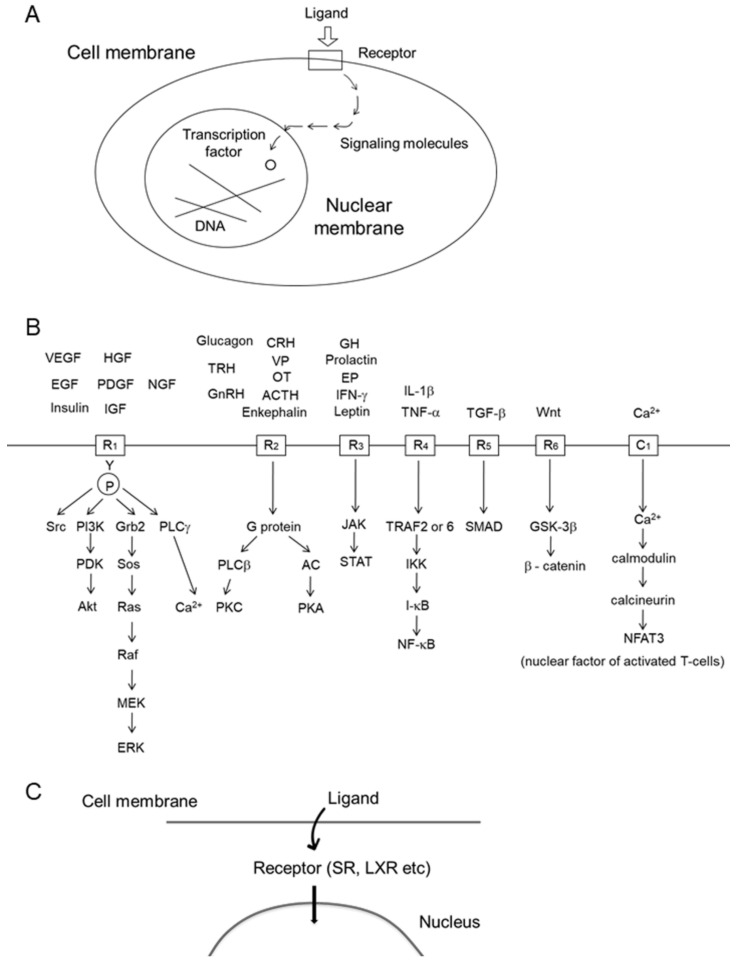
Cellular signal transduction pathways. (**A**) Image of intracellular signal transduction. (**B**) Typical signal transduction pathways commonly used in the cell. (**C**) Intracellular signaling via cytoplasmic receptors. SR, steroid receptor; LXR, liver X receptor.

**Figure 2 marinedrugs-19-00205-f002:**
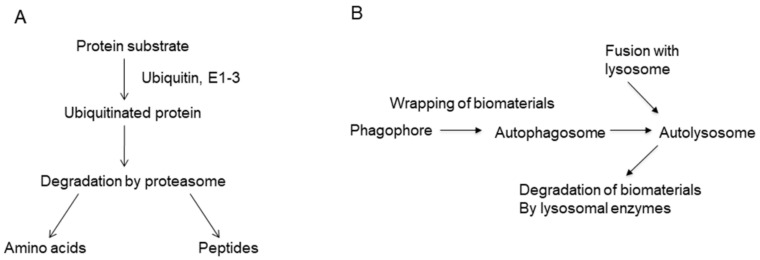
Protein degradation systems. (**A**) Ubiquitin–proteasome biodegradation system. (**B**) Process of autophagy.

**Figure 3 marinedrugs-19-00205-f003:**
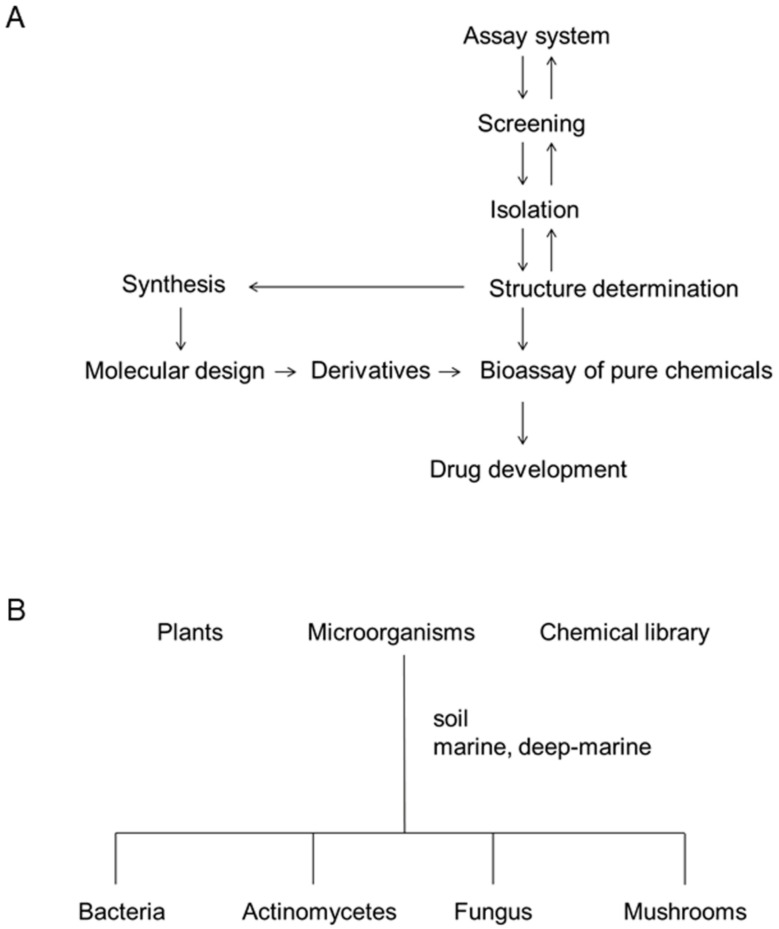
Screening of bioactive metabolites. (**A**) Process of natural product screening. (**B**) Screening sources.

**Figure 4 marinedrugs-19-00205-f004:**
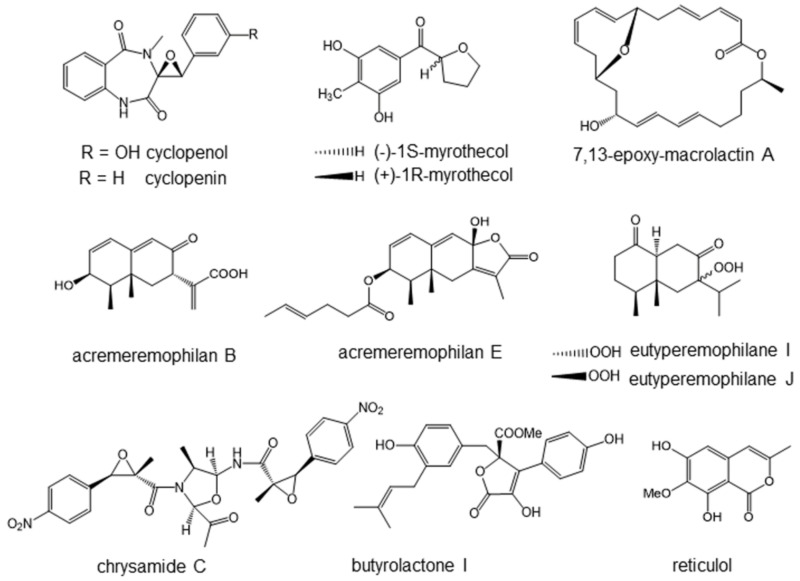
Anti-inflammatory agents isolated from deep-sea organisms.

**Figure 5 marinedrugs-19-00205-f005:**
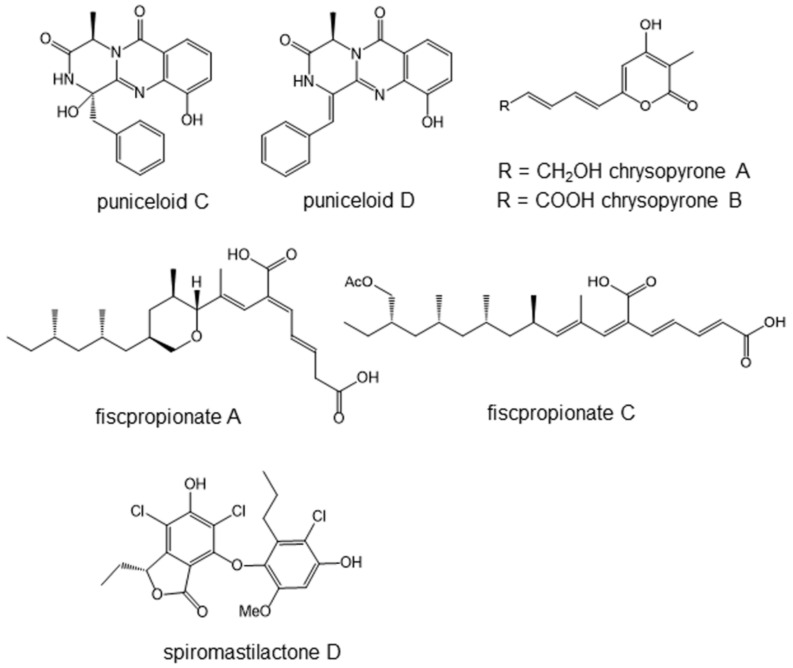
Modulators of metabolic syndrome and antimicrobial compounds isolated from deep-sea organisms.

**Figure 6 marinedrugs-19-00205-f006:**
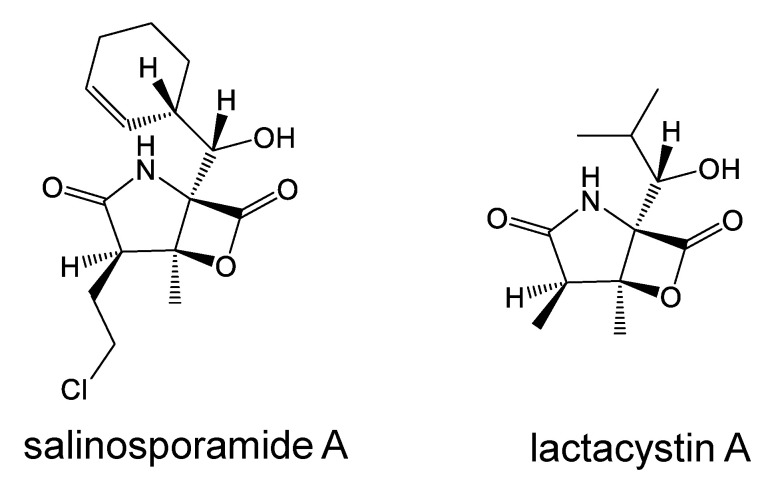
Proteasome inhibitors; Salinosporamide A shows anticancer activity.

**Table 1 marinedrugs-19-00205-t001:** Cell signaling inhibitors isolated from deep-sea organisms.

Natural Products	Target Signal	Related Illustration	Reference
Cyclopenol and cyclopenin	NO production (NF-κB)	Figure 1B, R_4_	[23]
Myrothecols	NO production	Figure 1B, R_4_	[26]
7,13-Epoxyl-macrolactin A	NO production	Figure 1B, R_4_	[27]
Acremeremophilane B	NO production	Figure 1B, R_4_	[28]
Eutyperemophilane I and J	NO production	Figure 1B, R_4_	[29]
Chrysamide C	IL-17 Production	__ *	[30]
Butyrolactone I	Mast cell activity	__	[34]
Reticurol	Mast cell activity	__	[35]
Puniceloids C and D	Liver X receptor	Figure 1C	[41]
Chrysopyrones A and B	Protein–tyrosine phosphatase	Figure 1B, R_1_	[43]
Fiscpropionate A and C	Protein–tyrosine phosphatase (bacterial)	__	[46]
Spiromastilactone D	Influenza virus	__	[47]
Salinosporamide A	Proteasome	Figure 2A	[52]

* The mechanism is unknown.

## Data Availability

No new data were created or analyzed in this study. Data sharing is not applicable to this article.

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
