# Peer review of "Cellular Signal Transductions and Their Inhibitors Derived from Deep-Sea Organisms"

_marinedrugs, 2021, doi:10.3390/md19040205_

Round 1

Reviewer 1 Report

I have reviewed the article and the article can be accepted in the present form.

Reviewer 2 Report

Whilst this is a very comprehensive review of signal transduction inhibitors derived from deep-sea organisms, I think there are a couple of significant issues that need to be addressed.

The main issue is that the section on anticancer agents doesn't speak to signal transduction inhibition, but mainly general cytotoxicity, and I think either the section should be revised to address signal transduction, or the cancer part should be removed.

The introductory section on signal transduction is not an easy read and flows poorly. I think this would benefit by addition of diagrams showing pathways and the points in the pathways at which the inhibitors mentioned in the text act. Also subdividing this section into smaller sub-sections would help.

Addition of summary tables covering the isolated agents and their applications would also be helpful in improving the structure and flow of the review.

Reviewer 3 Report

The manuscript by Wang and Umezawa is a Review reporting a survey of the effects produced by several compounds isolated from deep-sea organisms on cellular signal transduction processes whose alteration is involved in different diseases including cancer. The Authors conclude that deep-sea organisms represent emerging sources for the isolation of bioactive compounds for the development of signal transduction modulators useful for molecular medicine.

Minor comments

Page 2

Line 56: delete the phrase “We isolated a novel protein-tyrosine kinase inhibitor, lavendustin, from Streptomyces” leaving the reference.

Paragraphs 5 and 6

Except for lactacystin and salinosporamide A, the indication of all the compounds reported in these two paragraphs have been inverted between Figure 5 and 6. Please, carefully check the indication.

Figure 5

To render the illustration more comprehensive, move the structure of aphidicolin A8 on the same line of Ilamycin E1 and E2. The name of these latter has to be corrected.
